# Role of Percutaneous Endoscopic Gastrostomy for the Nutrition of Head and Neck Cancer Patients before and up to 6 Months after Cancer Treatment

**DOI:** 10.3390/cancers16183138

**Published:** 2024-09-12

**Authors:** Mussab Kouka, Sophie Brand, Sven Koscielny, Thomas Bitter, Klaus Pietschmann, Thomas Ernst, Orlando Guntinas-Lichius

**Affiliations:** 1Department of Otorhinolaryngology, Jena University Hospital, 07747 Jena, Germany; mussab.kouka@med.uni-jena.de (M.K.); sophie.brand@uni-jena.de (S.B.); sven.koscielny@med.uni-jena.de (S.K.); thomas.bitter@med.uni-jena.de (T.B.); 2Department of Radiation Oncology, Jena University Hospital, 07747 Jena, Germany; klaus.pietschmann@med.uni-jena.de; 3University Tumor Center, Jena University Hospital, 07747 Jena, Germany; thomas.ernst@med.uni-jena.de

**Keywords:** percutaneous endoscopic gastrostomy, PEG placement, nutrition, weight loss, head and neck cancer

## Abstract

**Simple Summary:**

When we consider nutrition in patients with head and neck cancer (HNC), we mostly focus on the type of nutrition; studies on the type of nutrition at different time points are sparse. This retrospective study analyzed patients with HNC according to their nutritional status and association with percutaneous endoscopic gastrostomy (PEG) from admission to six months after treatment at a tertiary hospital in Germany from 2017 to 2019. Results showed that 14.9% required PEG before starting therapy. The need for PEG increased to 22.7% at six weeks after therapy and remained stable at 23% six months later. PEG placement was more frequently required for alcohol or nicotine use, oropharyngeal and hypopharyngeal carcinoma, squamous cell carcinoma, cancer stage III/IV, chemotherapy and impaired feeding. There has been an increase in the percentage of PEG over the observation period.

**Abstract:**

This retrospective monocentric cohort study analyzed patients with head and neck cancer according their nutritional status and association of percutaneous endoscopic gastrostomy (PEG) from admission to six months after treatment at a tertiary hospital in Germany from 2017 to 2019. A total of 289 patients (76.5% men; median age 62 years; 63.3% stage IV) were included. Univariate analyses and ANOVAs with repeated measures were performed to analyze differences over time. The percentage of patients requiring PEG was 14.9% (43 of 289 patients) before start of treatment (Z0), 14% (40 of 286 patients alive) after one week (Z1), 22.7% (58 of 255 patients) after six weeks (Z2) and 23% (53 of 230 patients) after six months (Z3) from the end of treatment. PEG placement was associated with alcohol or nicotine consumption, in oropharyngeal and hypopharyngeal carcinoma, squamous cell carcinoma, cancer stage III/IV, chemotherapy and impairment of food intake (all *p* < 0.05). Weight loss between Z1 and Z3 with PEG did not differ from patients without PEG at Z0 (*p* = 0.074), although patients with PEG at Z0 had a lower mean weight at the beginning. PEG was important for a quarter of the patients alive at Z3 and helped to prevent weight loss.

## 1. Introduction

Maintaining or ameliorating the nutritional status is an important pillar in the treatment of head and neck cancer (HNC), as it has been proven that patients with HNC often suffer considerable weight loss before, during and after treatment [1,2,3]. HNC affects the most fundamental activities of the upper aerodigestive tract including speaking, breathing, eating and drinking [4]. Many patients with HNC already have a restricted nutritional status before or at the time of diagnosis. In addition, a premorbid lifestyle such as alcohol and nicotine abuse favors malnutrition [5]. This also has an impact on the prognosis of patients with HNC [2,6,7]. Furthermore, HNC is often the cause of dysphagia, odynophagia and taste disorders [4]. The prevalence of malnutrition is associated with the location of the tumor [6]. Malnutrition therefore plays a major role in patients with HNC [8]. In addition, cancer patients frequently suffer from metabolic changes attributable to systemic inflammation. These include increased energy requirements, proteolysis and lipolysis [9]. Ultimately, treatment of HNC exacerbates the problem of malnutrition [10]. Treatment-related toxicities include progressive dysphagia due to oral mucositis which occurs primarily during radiochemotherapy treatment [6,11].

As soon as the oral form of nutrition is no longer sufficient, an enteral form of nutritional support is favored over the parenteral form [12,13,14]. There are several ways to provide enteral nutrition. These include nasogastric (NG) and nasojejunal tubes as well as percutaneous endoscopic gastrostomy (PEG). On the one hand, the duration of the planned enteral nutrition is important. On the other hand, the patient’s wishes need to be taken into account, and the tumor’s characteristics and the type of treatment also need to be considered [15]. PEG has a special role in this context, as enteral nutrition can be provided over a longer period of time, as in the case of HNC patients who suffer from dysphagia and experience an exacerbation of symptoms during treatment. The advantage of PEG over other forms of enteral nutrition is the bypass of the upper aerodigestive tract. In addition, PEG is also indicated for patients with HNC in a palliative situation as they suffer from stenosis in the natural path of food due to the tumor [16]. However, there is discussion in the literature as to whether a prophylactic PEG should be placed if there is no swallowing disorder and no weight loss prior to treatment, or whether a reactive PEG should be placed if there is an existing swallowing disorder and malnutrition [17,18]. On the one hand, PEG placement can lead to acute (wound infection, bleeding, pneumoperitoneum, aspiration pneumonia and ileus) or chronic complications (tube leakage or obstruction, spontaneous tube removal, ulceration and necrosis of the stomach wall and recurrent aspiration), which are associated with a higher mortality rate; on the other hand, the swallowing muscles are not trained and scarring with stenosis develops more quickly, leading to more swallowing problems in the long term [19].

The nutritional status during the course of patients with HNC has not yet been comprehensively investigated in literature, especially the nutritional status several months after the end of treatment. In order to evaluate the nutritional situation and in particular PEG in a functional and oncological context, data from patients with HNC at a tertiary university hospital were analyzed.

## 2. Methods

### 2.1. Ethical Considerations

This monocentric, retrospective study was approved by the Ethics Committee of the University Hospital of Jena (IRB No. 2023-2944). The Ethics Committee waived the requirement for informed consent from patients because the study was non-interventional, retrospective, and all data were analyzed anonymously.

### 2.2. Study Design

This retrospective study included all patients with primary diagnosis of head and cancer in the Department of Otorhinolaryngology, Jena University Hospital, Germany, from January 2017 to December 2019. These years were chosen to guarantee a sufficient follow-up time. In total, 457 patients were initially registered. All patients with the diagnosis of primary HNC in the period from January 2017 to December 2017 were included. Duplicates were excluded. Pathological stages of the tumors were recorded using the UICC-and TNM (8th edition) classifications [20]. Ultimately, 289 patients were included in this study. Patients’ medical records were reviewed. All relevant parameters (patient characteristics, histopathological characteristics, treatment characteristics, type of nutrition, symptoms and follow-up) were recorded anonymously in a database. The documentation focused on PEG as a type of nutrition, weight and size parameters over time in patients with HNC. PEG insertion was primarily indicated in cases of low initial weight, severe restrictions on oral nutrition and before planned chemotherapy or radiochemotherapy treatment.

### 2.3. Timeline and Treatment

Four points in time were defined for the assessment of the nutrition status: the day of hospitalization/before treatment (Z0), and then one week (Z1), six weeks (Z2) and six months (Z3) after the end of head and neck cancer treatment. One of the most important and simplest ways to implement a nutritional process is anthropometry. This primarily involved measuring height and weight, and using these data to calculate the body mass index (BMI). This information on height, weight and nutritional characteristics including patient, tumor and treatment characteristics was obtained from the tumor board and operative reports. Letters from departments of ENT and Radiation Oncology contained important information about follow-up at the previously defined times (Z0–Z3).

It is also useful to assess nutritional status in combination with performance status and quality of life. The Karnofsky performance status was used for classification. The Karnofsky performance status was also taken from the tumor board reports written by the treating physician. At each time point, type of nutrition and weight were documented and the BMI and Karnofsky performance status were determined. The medical history always included questions about food intake complaints and information about the current diet.

The decision in favor of PEG or a nasogastric tube was made in accordance with the German guidelines for the nutrition of patients with different types of HNC [21,22,23]. These generally state that artificial feeding, a nasogastric tube or PEG is indicated in the postoperative period. If it is to be expected that oral nutrition will no longer be possible for at least a few weeks, the PEG tube has proven to be safe and effective. If tube feeding (transnasal or transcutaneous) is necessary, the PEG tube should be preferred to a nasogastric tube if dysphagia is present or expected to persist. In uncomplicated cases, however, feeding with soft or liquid high calorie food under local and systemic analgesia is recommended, provided the patient does not aspirate. If these measures are not sufficient, nutrition must be provided via a PEG tube or nasogastric tube or parenterally. Prophylactic PEG placement is recommended if intensive radiotherapy or radiochemotherapy is planned [24]. However, the German guidelines also point out that the available literature shows no significant benefit of prophylactic PEG placement in terms of weight progression, quality of life or oncological outcome [25,26,27]. Furthermore, it is described that there is no evidence for the correct timing of PEG placement [28,29]. It is also recommended to replace a nasogastric tube with PEG if dysphagia and especially aspiration continue to occur after the healing phase and intensive swallowing training.

### 2.4. Statistical Analysis

SPSS Statistics Version 27.0 (IBM Deutschland GmbH, Ehningen, Germany) was used to carry out the statistical analyses. Nominal and ordinal data are presented as absolute numbers and percentages. Metric data were calculated as mean, standard deviation, median and range. To compare nominal or ordinal data of two subgroups, the Pearson chi-square test or the Fischer exact test was used. The Mann–Whitney U test for two independent samples was used to test for differences of metric data of two subgroups. The significance level was set to *p* ≤ 0.05. All significant factors from these univariate analyses were included in a time-dependent analysis of variance (ANOVA) for the four primary outcome parameters (Z0, Z1, Z2 and Z3). This was preceded by a normality test according to Kolmogorov–Smirnov for the factors used in the ANOVAs. The rate of missing data was less than 5% per parameter. Based on the assumption that the missingness that occurred are completely random, the listwise deletion approach was used to handle the missing data in the ANOVA calculations. Due to the Mauchly’s test of sphericity, the sphericity assumption was violated in all ANOVA calculations. Therefore, an adjustment with the Huynh–Feldt correction was performed. Hence, all ANOVA data presented (mean square, F, *p*-values) include these corrections. The significance level was set to *p* ≤ 0.05.

## 3. Results

### 3.1. Patient’s Characteristics, Tumor Characteristics and Treatment Characteristics

The characteristics of the patients are shown in Table 1. A total of 289 patients were included. Of these, 221 (76.5%) were male and 68 (23.5%) were female. At the time of diagnosis, the mean age was 63 ± 13 years. Active or former alcohol consumption was present in over 40% (117 patients). Approximately two-thirds of the patients (66.1%) were active or former smokers.

Most tumors were located in the oropharynx (22.5%), larynx (15.2%) and oral cavity (18.7%). One-third of patients already had a T4 classification (31.8%) at the time of initial diagnosis. The distribution of the T1 classification (17.6%), T2 classification (15.6%) and T3 classification (20.1%) were broadly similar. At the time of diagnosis, no lymph node metastases (N0) were found in 40.8% of patients. The majority of patients had no distant metastases (M0; 229 patients, 79%); distant metastases (M+) were detected in 51 patients (17.6%). According to the UICC classification, most patients (32.9%) had stage IVa at the time of diagnosis. The treatment characteristics are shown in Table 2. In the majority of patients, surgery was chosen as the initial treatment, followed by radiotherapy or radiochemotherapy in almost a third of cases. After the last surgery, an R0 situation was achieved in 38.8%.

### 3.2. Nutrition and Feeding

Nutrition characteristics are listed in Table 3. The vast majority of patients already had a restored dental status at Z0 (68.2%). Only 15.2% of the patients had a tooth status that required restoration. In 41 patients (11.1%), a pre-therapeutical PEG tube had been placed at Z0. Nine of these patients had undergone surgery and thirty-two had not received surgical treatment. Hereinafter, more than three-quarters of the patients had not undergone surgery. Of the patients who did not receive surgical treatment, 18 patients (6.2%) did not receive a PEG. Only four patients (1.4%) received parenteral nutrition and seven patients (2.4%) received temporary suspension of food. The mean size of the patients was 173.11 ± 8.14 cm with a mean weight of 77.70 ± 17.32 kg, resulting in a BMI of 25.87 ± 5.09. Oral food intake was possible in 154 patients (53.3%) postoperatively. In 63 patients (21.8%), postoperative nutrition was provided via a nasogastric tube. The ways of feeding at the different points in time (Z1–Z3) are shown in Figure 1.

Tumor-related symptoms influencing nutrition are listed in Appendix A. The majority of patients reported various non-specific symptoms caused by the tumor. In about half of the cases, there was pain during nutrition, a globus pharyngeus or visible edema in the area of the tumor. Dysphagia was reported in 36.0% of cases and hoarseness in 23.5%. The mean number of symptoms number was 2.79 ± 1.50. Long-term results and follow-up measures are listed in Appendix A. In addition, Appendix A shows the symptoms reported by the patients, the forms of nutrition and the clinically measurable parameters at the three follow-up time points.

### 3.3. Distribution and Univariate Analysis of Associations on PEG

The results of the univariate analysis at the different time points (Z0, Z1, Z2 and Z3) are shown in Table 4 and Table 5. At Z0, patients with reported alcohol (*p* < 0.001) and/or nicotine consumption (*p* = 0.002) were statistically more likely to receive a PEG tube. All symptoms with physical impairment of food intake (dysphagia, pain during nutrition, gag reflex and globus pharyngeus) were reasons for the insertion of a PEG tube before the start of therapy (all *p* < 0.05). Unrestored dental status was associated with significant PEG placement (*p* = 0.008). With regard to the age of the patients, PEG tubes were increasingly placed in younger patients before the start of therapy (*p* = 0.015). The mean age of patients with a PEG tube prior to treatment was 59.62 ± 9.12 years. This compares to a mean age of 63.64 ± 12.86 years for patients without a PEG tube placed before the start of therapy. Weight and BMI were decisive parameters for the placement of a PEG tube at Z0. Patients with a PEG tube placed before the start of therapy were on average 10 kg lighter than patients without PEG (*p* = 0.001). This difference was therefore also reflected in the BMI (*p* < 0.001). The location of the tumor was also significant. Tumors in the direct path of food (oral cavity, oropharynx or hypopharynx) (*p* < 0.001) were a frequent reason for placement of a PEG at Z0. Furthermore, tumors in the higher stages (III and IV) and squamous cell carcinomas were predisposed to PEG insertion at Z0. In patients who had not reached R0 and in patients who had received chemotherapy, PEG was used more frequently (*p* < 0.001). However, as soon as a patient could be treated with surgery, they were less likely to receive a PEG preoperatively (*p* < 0.001).

At Z2, all parameters (with the exception of taste disturbances) were significantly related to PEG placement before the start of treatment. Patients who were able to feed themselves orally at Z2 were statistically less likely to receive PEG before the start of treatment (*p* < 0.001). Most patients who received a PEG tube before the start of therapy still had it at Z2 (*p* < 0.001). With regard to weight, BMI and the Karnofsky performance status at Z2, there were significant differences between the patient groups with and without a PEG placement before the start of treatment (*p* < 0.001). Patients with PEG weighed significantly less than patients in the other group at Z2. There was an average difference of 9% in Karnofsky performance status between the groups.

At Z3, significantly more patients with a PEG tube in place before the start of therapy had a swallowing disorder (*p* < 0.001). The symptom “pain during nutrition” was marginally significant compared to PEG placement before the start of therapy (*p* = 0.048). Patients who could be fed orally were significantly less likely to have a PEG tube placed before the start of therapy (*p* < 0.001). In most cases, possession of a PEG tube at Z3 required the PEG to be in place before the start of therapy (*p* < 0.001). The difference in mean weight between the two groups (with and without PEG before the start of therapy) was less significant at Z3 than at previous time points (*p* = 0.012). However, there was a difference in weight of around 9 kg. The BMI deviation and the Karnofsky performance status were both significant at Z3 (*p* < 0.001).

### 3.4. Time-Dependent Multivariate Analyses for the Association of Use of a PEG

Table 6 and Table 7 compare patients with and without PEG placement before the start of treatment with regard to various parameters over the course of the follow-up time. The results of the normality test for weight, BMI and Karnofsky performance status data are shown in the Appendix A. Weight and BMI data were normally distributed but Karnofsky performance status data were non-normal. For the entire cohort, all patients showed a significant mean weight loss of 4.85 kg over time (*p* < 0.001). Although patients with PEG placement before the start of treatment had a lower mean weight at the beginning, the weight loss over the course of the follow-up did not differ from the weight loss of patients without PEG placement before the start of treatment (*p* = 0.074). For the BMI, the same was observed. There was a significant reduction in BMI in the entire cohort (*p* < 0.001), but there were no differences in BMI reduction between patients with and without PEG placement before the start of treatment (*p* = 0.079). When looking at the Karnofsky performance status of the entire cohort, a weakly significant reduction was observed during the follow-up (*p* = 0.047). There were no significant differences in the reduction of the Karnofsky performance status between patients with and without PEG placement before the start of therapy (*p* = 0.997). However, patients without PEG placement before the start of treatment initially had higher percentages of Karnofsky performance status.

## 4. Discussion

In the present study, PEG was evaluated in the nutritional status of patients with HNC. PEG placement was associated with certain parameters such as alcohol or nicotine consumption at Z0, localization in oropharyngeal and hypopharyngeal carcinoma at Z0, squamous cell carcinoma at Z0, higher cancer stages (stage III/IV) at Z0, chemotherapy and impairment of food intake (swallowing disorders, pain during nutrition and globus pharyngeus) at Z0–Z3. The percentage of patients requiring PEG increased over the course of the follow-up. The percentage values were 14.9% (43 of 289 patients) before the start of treatment (Z0), 14% (40 of 286 patients) at Z1, 22.7% (58 of 255 patients) at Z2 and finally 23% (53 of 230 patients) at Z3. It has to be emphasized when interpreting the relative numbers that the number of patients alive decreased during the follow-up. In the study by Wermker et al., which compared a PEG-dependent group with a control group, the percentage of PEG was 17% (without specifying the exact date) [30]. In a retrospective study of 196 patients with HNC, Nugent et al. showed a percentage of PEG of 22.5% (again without specifying the exact date). These results are also comparable to those of the present study [31]. The study by Nugent et al. did not analyze different time points, but only compared prophylactic PEG placement (P-PEG) with therapeutic PEG placed during treatment (T-PEG). In the present study, 14% of patients received a prophylactic PEG. The placement of a PEG before the start of treatment as a prophylactic PEG is discussed in the literature [32,33,34]. Din-Lovinescu et al. analyzed 4068 cases with HNC from the Nationwide Inpatient Sample (NIS) from 2003 to 2014 from US non-federal hospitals and reported a significant benefit of prophylactic PEG in terms of lower complication rates, shorter length of stay and lower hospital costs [34]. In this study of NIS data, most patients had a tumor in the digastric tract, in the oral cavity (prophylactic PEG 37.6%; late PEG 35.2%) and in the oropharynx (prophylactic PEG 25.6%; late PEG 24.7%). However, only radiotherapy was taken into account for treatment. Also of note is the randomized, controlled phase III trial, the Swall-PEG study by Dragan et al. with the primary endpoint of patient-reported outcomes in terms of swallowing and quality of life after prophylactic versus reactive PEG tube placement in advanced oropharyngeal cancer patients treated with definitive chemoradiotherapy [35]. The results have not yet been published.

In our study, the indication for PEG was increasingly given for certain localizations. These were tumors of the hypopharynx and oropharynx. The study by Zuercher et al. showed a correlation between these two localizations and also between tumors of the oral cavity and PEG. These three localizations were thus associated with more than three-quarters of PEG placement [36]. The study by Sieron et al. also confirmed these two localizations as predisposing to prophylactic PEG [37]. In Riera et al., hypo- and oropharyngeal carcinomas were among the most common sites for PEG placement, but laryngeal carcinomas showed a stronger association with PEG use than HNC of hypopharynx and oropharynx [38]. It is possible that the smaller number of laryngeal cancer cases in the present study had an influence on the fact that there was no significance between laryngeal cancer and PEG use. There was still a dependency on the cancer stage. Patients with a higher stage required PEG significantly more often than patients with a lower stage. This was also found to be the case in the studies by Zuercher et al., Riera et al. and Wermker et al. [30,36,38]. Riera et al. also demonstrated that squamous cell carcinomas were more frequently associated with PEG use [38]. This was also found by Sieron et al. for prophylactic PEG use in squamous cell carcinoma, which is consistent with our results [37]. Current or previous nicotine and/or alcohol consumption were also predictive factors for the placement of a PEG tube and for the placement of a prophylactic PEG tube which is confirmed by Wermker et al. [30]. PEG placement was often indicated for all types of complaints relating to oral feeding. Riera et al. found dysphagia to be the most common indication for PEG insertion. Wermker et al. showed similar results.

With regard to the association between the treatment and PEG placement, Zuercher et al. described that patients with chemotherapy, radiotherapy or radiochemotherapy primarily required a PEG. In our study, we were able to show that chemotherapy or radiochemotherapy is associated with PEG placement. The study by Sieron et al. showed similar results. Van der Linden et al. also added that patients who received radiotherapy alone were more likely to receive a prophylactic PEG. It was argued that radiotherapy may worsen existing poor dental status and that poor dental status was associated with PEG placement. Wermker et al. were also able to establish this. Furthermore, patients who were primarily treated with surgery were significantly less likely to require a PEG tube before starting treatment [39]. The present study was also able to show the same results. Ehrsson et al. stated an average PEG duration of 3 to 12 months. Accordingly, it was confirmed that patients with a PEG tube at the previous follow-up were very likely to still have it at the next follow-up [40], which is consistent with our results. Patients with a prophylactic PEG used it for longer than patients who only received the PEG tube after completion of treatment [41]. Nguyen et al. stated a mean duration of 8 months until a prophylactic PEG was removed [42]. This also explains that patients with prophylactic PEG were still using the PEG more frequently at the follow-up (Z1, Z2 and Z3) than patients with subsequently placed PEG. It is possible that the PEG dependency is the reason why patients with a prophylactic PEG were significantly less able to feed themselves orally at the individual follow-up than patients without a prophylactic PEG tube.

There were significant differences in weight and BMI between patients with and without PEG. Patients with PEG showed a lower weight and thus also a lower BMI than patients without PEG. Wermker et al. were also able to demonstrate the same results, but the Karnofsky performance status was not affected by these significant differences, as was the case in our study. In their retrospective study of 186 patients with HNC who received a PEG placement prior to treatment and underwent radiotherapy or chemoradiotherapy, Lang et al. categorized all patients into three groups according to their weight and BMI [43]. Group 2 (BMI/weight range 12–18/33 kg–63 kg), which is most comparable to the patients in our study, showed only a slight change in weight (initial weight: 66.5 kg; weight at the end of treatment: 65 kg) and BMI (initial BMI: 22.1, BMI at the end of treatment: 21.6). The ANOVA in our study showed that with regard to the group differences between patients with and without prophylactic PEG, there were no differences in weight loss and in reduction in Karnofsky performance status between the two groups. It was only recognizable that patients with PEG had a lower initial weight than patients without PEG and had higher initial values in Karnofsky performance status. Patients with PEG had an average weight loss of 5.48 kg. This corresponds to a percentage value of 7.6%. In the study by Nugent et al., the total weight loss of patients with PEG was only 5.2%. However, it should be noted that in Nugent et al., the patients with prophylactic PEG showed a much lower weight loss than the patients with subsequently placed PEG. The values are slightly distorted due to these differences in weight loss. In Nugent et al., the percentage of weight loss would be 7.5% if only PEG tubes placed during or after therapy were considered. This value is also comparable with the results of our study [31]. In summary, the percentage weight loss of 7.6% is within the expected 6–12% for patients with HNC. Therefore, PEG feeding is considered acceptable. Regarding the quality of life, it was found that patients with PEG tubes reported more frequent discomfort with food intake. In addition, these patients showed greater weight loss. It could therefore be assumed that patients with a PEG tube had a lower quality of life than patients with oral nutrition. Low weight loss also had a positive effect on quality of life. As patients with oral nutrition showed the least weight loss in this study, it can be assumed that these patients had the highest quality of life. Prevost et al. also found that a better nutritional status correlates with a better quality of life [44]. Ehrsson et al. described that patients with enteral nutrition had a lower quality of life than patients with oral nutrition. The patients found the enteral feeding tubes irritating and missed oral nutrition [45].

The decision of who indicates/decides to insert a PEG tube varies considerably depending on the institution. To this end, the management of tube feeding in an interdisciplinary context in Germany was analyzed in a web-based survey by seventy participants (forty-two radiation oncologists, twelve medical oncologists, fourteen head and neck surgeons and two physicians from several specialties) in a study by Löser et al. [46]. The study showed that a significant proportion of participants (70%) do not perform standardized nutritional screening prior to a planned chemoradiation and that there are significant differences between institutions and specialties in Germany. However, several studies have already confirmed the benefits of pre-therapeutic nutritional screening [47,48]. Löser et al. cited work overload and staff shortages in the German healthcare system as possible reasons for not carrying out pre-therapeutic nutritional screening. Furthermore, Löser et al. stated that 80% of respondents were in favor of prophylactic PEG tube insertion. However, it should be noted that radiation oncologists are overrepresented in this web-based study.

The present study was limited by a retrospective monocentric study design. As a result, decisions regarding nutritional treatment could not be retraced retrospectively. Thus, associations could be analyzed on the basis of the available data, but no causal relationships could be determined. Furthermore, missing values, although this was rare, could have led to a selection bias. In addition, there were also incomplete data sets for instances like socioeconomic factors or the time of death. Furthermore, the impact of PEG tube use on overall survival is an important factor but was not the focus of the present study. We are currently conducting a study to investigate the impact of PEG tube use on survival. Although nutritional risk screening scores typically do not include blood values due to high variability [49], some factors like albumin seem to be surrogate markers for the nutritional status of a head and neck cancer patient [50]. Hence, it would be worthwhile to include metabolic blood values in further investigations.

Overall, the present study was able to analyze PEG for the nutrition of patients with HNC before treatment and after several follow-ups. This is in contrast to many studies in the literature. There is often an analysis of just one point time before and after treatment. The patient data were collected from a tertiary university hospital. A prospective study in a larger multicenter setting could provide further and more precise information on the nutritional status in HNC. In any case, nutritional aspects should be included in the therapeutic measures for HNC.

## 5. Conclusions

This retrospective study from a tertiary university hospital in Thuringia from 2017 to 2019 describes the placement and use of percutaneous endoscopic gastrostomy (PEG) in the nutritional status of 289 patients with head and neck cancer (HNC) from admission to six months after treatment at a tertiary hospital. PEG tube placement and PEG tube use were favored by alcohol or nicotine abuses, localization in oropharyngeal and hypopharyngeal carcinoma, squamous cell carcinoma, higher cancer stages (stage III/IV), chemotherapy and impairment of food intake (swallowing disorders, pain during nutrition, gag reflex and globus pharyngeus). PEG tube placement was important for almost a quarter of the patients (53 of 230 alive patients) at the final follow-up six months after treatment.

## Figures and Tables

**Figure 1 cancers-16-03138-f001:**
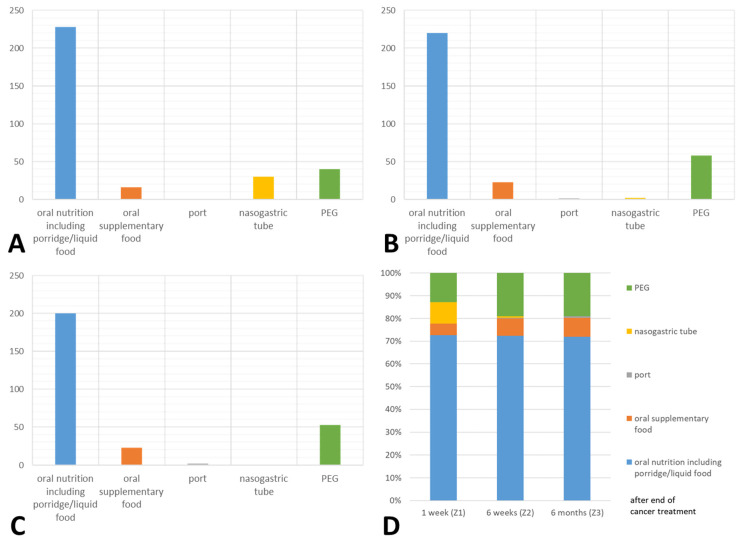
Frequency distribution of the applied types of nutrition after one week (Z1) (**A**); after six weeks (Z2) (**B**); and after six months (Z3) (**C**) from the last treatment; as well as the percentage distribution of the different nutrition types over the time (**D**). PEG = percutaneous endoscopic gastrostomy; port = totally implantable venous access device.

**Table 1 cancers-16-03138-t001:** Patients’ characteristics and tumor characteristics.

Parameter	Frequency (*n*)	%
All	289	100
Gender		
Male	221	76.5
Female	68	23.5
Year of diagnosis		
2017	101	34.9
2018	93	32.2
2019	95	32.9
Cigarette smoking		
Yes	137	47.4
No	98	33.9
Former	54	18.7
Alcohol drinking		
Yes	86	29.8
No	172	59.5
Former	31	10.7
Tumor localization		
Oral cavity	54	18.7
Hypopharynx	35	12.1
Oropharynx	65	22.5
Nasopharynx	4	1.4
Larynx	44	15.2
Salivary glands	35	12.1
Nose and paranasal sinus	13	4.5
Thyroid	10	3.5
Others	29	10.0
Histology		
Squamous cell carcinoma	229	79.2
No squamous cell carcinoma	60	20.8
p16		
positive	33	11.4
negative	169	58.5
Not analyzed	87	30.1
T classification		
T0	28	9.7
Tis	7	2.4
T1	51	17.6
T2	45	15.6
T3	58	20.1
T4	92	31.8
TX	8	2.8
N classification		
N0	118	40.8
N1	41	14.2
N2	77	26.6
N3	45	15.6
NX	8	2.7
M classification		
M0	229	79.2
M1	51	17.6
MX	9	3.2
UICC Staging		
Stage 0	6	2.1
Stage I	33	11.4
Stage II	22	7.6
Stage III	41	14.2
Stage IVa	95	32.9
Stage IVb	39	13.5
Stage IVc	49	16.9
Stage unknown	4	1.4
Residual tumor		
R0	167	57.8
R1	12	4.2
R2	14	4.8
Rx	23	8.0
No surgery performed	73	25.2
Grading		
G1	17	5.9
G2	153	52.9
G3	55	19.0
G4	2	0.6
GX	62	21.6

**Table 2 cancers-16-03138-t002:** Treatment characteristics.

Parameter	Frequency (*n*)	%
All	289	100
Radiotherapy	8	2.8
Chemotherapy	11	3.8
Surgery alone	100	34.6
Surgery and postoperative adjuvant treatment		
Radiotherapy	57	19.7
Chemotherapy	4	1.4
Radiochemotherapy	58	20.1
Neck dissection	166	57.5
Primary radiochemotherapy	33	11.4
Best supportive care	12	4.2
No treatment	4	1.3
Neoadjuvant chemotherapy and definitive therapy	2	0.7
Number of operations performed		
0	11	3.8
1	125	43.3
2	117	40.5
3	29	10.0
>3	7	2.4
R classification after the last surgery		
R0	112	38.8
R1	7	2.4
R2	7	2.4
Rx	17	5.9
Inoperable	10	3.5
Not applicable, as not operated	136	47.0
	Mean ± SD	Median, Range
Duration of inpatient stay for surgical treatment in days	22.0 ± 21.31	16; 0–189
Duration until the first operation after inpatient admission (N = 281)	1.02 ± 1.21	1; 0–7
Duration of chemotherapy in days (N = 106)	63.07 ± 57.41	44; 1–279
Duration of radiotherapy in days (N = 153)	43.64 ± 13.67	44; 1–98

SD = standard deviation.

**Table 3 cancers-16-03138-t003:** Nutrition characteristics before and after starting treatment.

Parameter	Frequency (*n*)	%
Dental status		
Restored	197	68.2
In need of refurbishment	44	15.2
Partially refurbished	24	8.3
Toothless	15	5.2
Not specified	9	3.1
PEG applied		
Yes	41	14.2
Of which received an operation in the course of therapy	9	3.1
Of which no surgery received in the course of therapy	32	11.1
No	248	85.8
Nutritional status who have not undergone surgery		
PEG placed pre-therapeutically	32	11.1
no PEG placed pre-therapeutically	18	6.2
Parenteral	4	1.4
Temporary suspension of food	7	2.4
Postoperative nutrition		
Completely oral	154	53.3
Nasogastric tube	63	21.8
PEG postoperative placed	2	0.7
	Mean ± SD	Median, Range
Age in years	63 ± 13	62, 11–93
Size in cm (N = 271)	173.11 ± 8.14	174, 148–193
Weight in kg (N = 272)	77.70 ± 17.32	77, 41–145
BMI (N = 271)	25.87 ± 5.09	25.5, 15.4–43.3

SD = standard deviation; BMI = body mass index; PEG = percutaneous endoscopic gastrostomy.

**Table 4 cancers-16-03138-t004:** Univariate analysis of associations between PEG and patients’ characteristics, tumor characteristics and treatment characteristics before start of cancer treatment (Z0).

	Need of a PEG at Z0		
Parameter	Yes, *n*	No, *n*	*p*
Gender	41	248	0.293
Female	7	61	
Male	34	187	
Alcohol drinking	41	248	**<0.001**
Yes	28	89	
No	13	159	
Cigarette smoking	41	248	**0.002**
Yes	36	155	
No	5	93	
Tumor in the upper digastric tract (oral cavity, oropharynx or hypopharynx)	41	248	**<0.001**
Yes	37	117	
No	4	131	
Cancer stage	41	244	**0.001**
Stage I/II	1	60	
Stage III/IV	40	184	
Squamous cell carcinoma	41	248	**<0.001**
Yes	41	188	
No	0	60	
Grading	38	189	0.824
G1/2	29	141	
G3/4	9	48	
Residual tumor	39	236	**<0.001**
R0	7	160	
R+	32	76	
Surgery	41	248	**<0.001**
Yes	14	217	
No	27	31	
Chemotherapy	41	248	**<0.001**
Yes	30	76	
No	11	172	
Radiotherapy	41	248	0.074
Yes	27	126	
No	14	122	
Radiochemotherapy	41	248	**<0.001**
Yes	20	13	
No	21	235	
Postoperative complications	41	248	0.076
Yes	3	46	
No	38	202	

Significant *p*-values (*p* < 0.05) in bold.

**Table 5 cancers-16-03138-t005:** Univariate analysis of associations on PEG and various parameters at Z0, Z1, Z2 and Z3.

	Before Cancer Treatment (Z0)			*p*		1 Week after End of Cancer Treatment (Z1)			*p*		6 Weeks after End of Cancer Treatment (Z2)			*p*		6 Months after End of Cancer Treatment (Z3)			*p*	
Parameter	All, *n*	Yes, *n*	No, *n*			All, *n*	Yes, *n*	No, *n*			All, *n*	Yes, *n*	No, *n*			All, *n*	Yes, *n*	No, *n*		
Swallowing Disorders	289	41	248	**<0.001**		286	41	245	**<0.001**		255	33	222	**<0.001**		230	27	203	**<0.001**	
Yes	104	37	67			122	34	88			95	27	68			85	22	63		
No	185	4	181			164	7	157			160	6	154			145	5	140		
Pain during nutrition	289	41	248	**<0.001**		286	41	245	**<0.001**		255	33	222	**<0.001**		230	27	203	0.048	
Yes	150	37	113			151	36	115			74	20	54			51	10	41		
No	139	4	135			135	5	130			181	13	168			179	17	162		
Loss of appetite	289	41	248	0.205		286	41	245	**<0.001**		255	33	222	**<0.001**		230	27	203	0.167	
Yes	11	3	8			10	6	4			31	10	21			27	1	26		
No	278	38	240			276	35	241			224	23	201			203	26	177		
Gag reflex during nutrition	289	41	248	0.028		286	41	245	0.052		255	33	222	**<0.001**		230	27	203	0.144	
Yes	7	3	4			47	11	36			22	9	13			15	0	15		
No	282	38	244			239	30	209			233	24	209			215	27	188		
Taste disorders	289	41	248	0.564		286	41	245	0.476		255	33	222	0.597		230	27	203	0.324	
Yes	2	0	2			3	0	3			11	2	9			22	4	18		
No	287	41	246			283	41	242			244	31	213			208	23	185		
Globus pharyngeus	289	41	248	**<0.001**		286	41	245	**<0.001**		255	33	222	**<0.001**		230	27	203	0.143	
Yes	136	29	107			45	16	29			16	8	8			12	3	9		
No	153	12	141			241	25	216			239	25	214			218	24	194		
Restored dental status	280	40	240	0.008		279	40	239	0.001		250	32	218	0.022		228	27	201	0.011	
Yes	197	21	176			200	20	180			189	19	170			172	15	157		
No	83	19	64			79	20	59			61	13	48			56	12	44		
Swallowing therapy	289	-	-	-		286	41	245	0.019		255	33	222	0.868		230	27	203	0.952	
Yes	0	-	-	-		51	2	49			9	1	8			9	1	8		
No	289	-	-	-		235	39	196			246	32	214			221	26	195		
Oral nutrition including liquid food/puree	289	-	-	-		286	41	245	<0.001		255	33	222	**<0.001**		230	27	203	**<0.001**	
Yes	0	-	-	-		228	19	209			220	17	203			200	17	183		
No	289	-	-	-		58	22	36			35	16	19			30	10	20		
Oral supplementary food	289	-	-	-		286	41	245	0.604		255	33	222	**0.049**		230	27	203	0.838	
Yes	0	-	-	-		16	3	13			23	6	17			23	3	20		
No	289	-	-	-		270	38	232			232	27	205			207	24	183		
Nasogastric tube	289	-	-	-		286	41	245	0.018		253	33	222	**0.584**		230	27	203	-	
Yes	63	-	-	-		30	0	30			2	0	2			NA	0	NA		
No	2			-		256	41	215			253	33	220			NA	27	NA		
PEG	289	-	-	-		286	41	245	<0.001		255	33	222	**<0.001**		230	27	203	**<0.001**	
Yes	43	-	-	-		40	34	6			58	29	29			53	23	30		
No	246	-	-	-		246	7	239			197	4	193			177	4	173		
	PEG					PEG					PEG					PEG				
	Yes, *n* = 39		No, *n* = 246		*p*	Yes, *n* = 39		No, *n* = 230		*p*	Yes, *n* = 33		No, *n* = 246		*p*	Yes, *n* = 27		No, *n* = 224		*p*
	M ± SD	95% CI	M ± SD	95% CI		M ± SD	95% CI	M ± SD	95% CI		M ± SD	95% CI	M ± SD	95% CI		M ± SD	95% CI	M ± SD	95% CI	
Age in years	59.6 ± 9.12	56.7–62.6	63.6 ± 12.9	61.9–65.3	**0.015**	-	-	-	-	-	-	-	-			-	-	-	-	
Weight in kg	69.7 ± 15.4	64.8–74.7	79.1 ± 17.3	76.9–81.4	**0.001**	67.5 ± 14.6	62.8–72.3	78.1 ± 17.1	75.9–80.3	**<0.001**	66.5 ± 13.1	61.9–71.1	77.6 ± 17.2	75.2–79.9	<0.001	66.8 ± 11.9	62.1–71.5	75.5 ± 17.2	73.1–77.9	**0.012**
BMI	22.8 ± 4.45	21.4–24.2	26.4 ± 5.02	25.7–27.0	**<0.001**	22.17 ± 4.2	20.7–23.4	26.0 ± 5.0	25.4–26.7	**<0.001**	21.7 ± 4.0	20.3–23.1	25.8 ± 5.02	25.2–26.5	<0.001	21.8 ± 3.24	20.5–23.1	25.1 ± 4.90	24.4–25.8	**<0.001**
Karnofsky performance status in percent	NA	NA	NA	NA	NA	80.0 ± 12.7	75.9–84.1	88.4 ± 11.6	86.8–89.9	**<0.001**	78.2 ± 12.4	73.8–82.6	87.2 ± 10.4	85.8–88.6	<0.001	80.0 ± 9.61	76.2–83.8	87.3 ± 11.5	85.7–88.9	**<0.001**

M = mean; SD = standard deviation; BMI = body mass index; PEG = percutaneous endoscopic gastrostomy, NA = not applicable; day of hospitalization/before treatment (Z0), after one week (Z1), after six weeks (Z2) and after six months (Z3) from the last treatment; significant *p*-values (*p* < 0.05) in bold. At Z1, patients with a PEG tube placed before the start of therapy were more likely to have swallowing disorders, pain during nutrition, loss of appetite and a globus pharyngeus (all *p* < 0.001). More patients with a PEG tube placed before the start of therapy were no longer able to feed orally at Z1 (*p* < 0.001). No patient with a nasogastric tube at Z1 had received PEG (*p* = 0.018). The majority of patients with a PEG tube before the start of therapy still had it at Z1 (*p* < 0.001). A dental status in need of restoration was also an indicator for the placement of a PEG tube before the start of treatment (*p* = 0.001). At Z1, the differences between patients with and without PEG/PE placed before the start of treatment were statistically significant in terms of weight, BMI and Karnofsky performance status (*p* < 0.001 in each case). Patients without PEG were on average more than 10 kg heavier at Z1.

**Table 6 cancers-16-03138-t006:** Time-dependent multivariate ANOVA with repeated measures for the association between the placement of a PEG before the start of therapy compared to no PEG placement before the start of therapy and various clinical parameters.

Parameter	Square Sum	df	Mean of the Squares	F	*p*	Partial Eta-Square
Weight change between Z0 to Z3	1233.81	1.58	782.94	43.78	**<0.001**	0.167
Weight change in group comparison between Z0 to Z3	79.52	1.58	50.46	2.82	0.074	0.013
BMI change between Z0 to Z3	131.0	1.58	82.97	42.64	**<0.001**	0.163
BMI change in group comparison between Z0 to Z3	8.40	1.58	5.32	2.73	0.079	0.012
Karnofsky performance status change between Z1 to Z3	282.16	1.87	151.21	3.15	**0.047**	0.014
Karnofsky performance status change in group comparison between Z1 to Z3	0.158	1.87	0.085	0.002	0.997	<0.001

df = degrees of freedom; F = F-value; PEG = percutaneous endoscopic gastrostomy; BMI = body mass index; day of hospitalization/before treatment (Z0), after one week (Z1), after six weeks (Z2) and after six months (Z3) from the last treatment; significant *p*-values (*p* < 0.05) in bold.

**Table 7 cancers-16-03138-t007:** Comparison of various parameters in patients with and without a PEG in place before the start of treatment during the course of follow-up.

Parameter	PEG before Start of Treatment	Time Point	Mean	95% CI	
				Lower 95% CI	Upper 95% CI
Weight in kg	Yes (N = 27)	Z0	72.30	65.71	78.88
		Z1	69.44	62.98	75.91
		Z2	67.52	61.14	73.90
		Z3	66.82	60.49	73.14
	No (N = 194)	Z0	79.73	77.28	82.19
		Z1	78.58	76.17	80.99
		Z2	77.49	75.11	79.87
		Z3	75.51	73.15	77.87
	All (N = 221)	Z0	76.01	72.50	79.53
		Z1	74.01	70.56	77.46
		Z2	72.50	69.10	75.91
		Z3	71.16	67.79	74.54
BMI	Yes (N = 27)	Z0	23.55	21.67	25.42
		Z1	22.62	20.77	24.46
		Z2	22.00	20.18	23.81
		Z3	21.79	19.98	23.59
	No (N = 194)	Z0	26.51	25.81	27.21
		Z1	26.12	25.44	26.81
		Z2	25.79	25.10	26.46
		Z3	25.11	24.44	25.78
	All (N = 221)	Z0	25.03	24.03	26.03
		Z1	24.37	23.39	25.35
		Z2	23.89	22.92	24.86
		Z3	23.45	22.49	24.41
Karnofsky performance status in percent	Yes (N = 27)	Z1	82.22	78.21	86.23
		Z2	80.37	76.39	84.35
		Z3	80.00	75.65	84.36
	No (N = 199)	Z1	89.60	88.12	91.08
		Z2	87.64	86.17	89.11
		Z3	87.29	85.68	88.89
	All (N = 226)	Z1	85.91	83.77	88.05
		Z2	84.00	81.88	86.13
		Z3	83.64	81.32	85.96

PEG = percutaneous endoscopic gastrostomy; BMI = body mass index, day of hospitalization/before treatment (Z0), after one week (Z1), after six weeks (Z2) and after six months (Z3) from the last treatment.

## Data Availability

The original contributions presented in the study are included in the article and in the Appendix A. Further inquiries can be directed to the corresponding author.

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
