# Peer review of "Role of Percutaneous Endoscopic Gastrostomy for the Nutrition of Head and Neck Cancer Patients before and up to 6 Months after Cancer Treatment"

_cancers, 2024, doi:10.3390/cancers16183138_

Round 1

Reviewer 1 Report

Comments and Suggestions for Authors

To the Authors

Your manuscript is a highly detailed study on nutritional management during the treatment of HNSCC, with a particular focus on PEG. Although this is a single-center study with some limitations, it has significant implications for future research and analysis, potentially serving as a stepping stone for prospective studies in collaboration with other institutions. After careful consideration of your study, I would like to request the following substantial revisions to enhance the quality and clarity of your manuscript:

1.Concerns Regarding the ANOVA Statistical Method: Demonstrating the normality of the data is a crucial assumption for repeated measures ANOVA. Please provide the methods and results of the normality tests for the primary variables (e.g., weight, BMI, Karnofsky Performance Status) at each time point. Additionally, please evaluate sphericity. Moreover, the manuscript lacks a discussion on how missing data were handled in the ANOVA analysis, which needs to be clarified.

2.Clarification of General Treatment Guidelines: Your study focuses on preventive versus reactive PEG placement in head and neck cancer patients. However, the manuscript lacks a comprehensive description of the general treatment guidelines for head and neck cancer patients at your institution and in Germany. Specifically, PEG is considered a last resort, and nasogastric tubes or TPN are commonly used for acute nutritional management in some regions. You need to clarify why PEG was chosen and why TPN was not considered. This clarification is essential as it directly impacts decisions regarding pre- or post-treatment PEG placement. The rationale behind the clinical decision-making process for preventive versus reactive PEG should be more detailed.

3.Impact of PEG Placement on QOL: Your study suggests that PEG placement may lead to a decline in QOL. However, multiple studies have reported that proper nutritional management is crucial for maintaining QOL. This point requires a more detailed discussion and the introduction of counterevidence. Was the decline in QOL due to bias, issues in the treatment plan, or other factors?

4.Detailed Description of the Nutritional Screening Process: Currently, the manuscript lacks a detailed description of the nutritional screening process, including the specific procedures and criteria for KPS evaluation. It is important to clarify who conducted the KPS evaluations, at what stage, and whether these evaluations were consistent across different patients.

5.Impact of Alcohol and Nicotine Use: The manuscript briefly touches upon the impact of alcohol and nicotine use on nutritional status and treatment outcomes. However, further research is needed on this topic. Could you provide more detailed data on how these factors affected patient outcomes and whether subgroup analyses were performed, such as for patients who smoked enough to develop COPD or drank enough to develop alcohol-related liver disease?

Overall, this is a very interesting and important paper. I hope you can make it even better.

Sincerely,

Author Response

Point-by-Point Rebuttal Letter - Reviewer #1

Manuscript ID cancers-3184364

Role of percutaneous endoscopic gastrostomy for the nutrition of head and neck cancer patients before and up to 6 months after cancer treatment

We thank the reviewers for their helpful comments. We answer here all comments and queries point-by-point.

Reviewer #1

Your manuscript is a highly detailed study on nutritional management during the treatment of HNSCC, with a particular focus on PEG. Although this is a single-center study with some limitations, it has significant implications for future research and analysis, potentially serving as a stepping stone for prospective studies in collaboration with other institutions. After careful consideration of your study, I would like to request the following substantial revisions to enhance the quality and clarity of your manuscript

1.1. Concerns regarding the ANOVA statistical method: Demonstrating the normality of the data is a crucial assumption for repeated measures ANOVA. Please provide the methods and results of the normality tests for the primary variables (e.g., weight, BMI, Karnofsky Performance Status) at each time point. Additionally, please evaluate sphericity. Moreover, the manuscript lacks a discussion on how missing data were handled in the ANOVA analysis, which needs to be clarified.

Answer 1.1: Thank you for this important comment. Normality was tested now for all parameters included in the ANOVAs for Z0 to Z3, i.e. weight, BMI und Karnofsky. We have added a new supplementary table 4 with the results of the Kolmogorov-Smirnov normality test and a more detailed description in the Method section in line 145-146 and in the results section in line 259-260:

“In addition, a normality test according to Kolmogorov-Smirnov was calculated for the factors used in the ANOVA”

“The results of the normality test for weight, BMI and Karnofsky performance status are shown in the Supplementary Table 4. Weight and BMI data were normally distributed but Karnofsky performance status data were non-normal“

The Kolmogorov-Smirnov normality test did reveal normality for the weight and BMI data. There was no normal distribution of the Karnofsky data. The effect seems to be small, as the ANOVA tolerates such a small violations to its normality assumption rather well. The effect of the Karnofsky was anyway very small. The important predictors were weight and BMI. Hence, we did not take out the Karnofsky data form the calculations. We clearly address this now in the Results, lines 260-261, so that the reader can interpret the data correctly.

Concerning sphericity and missing values, we followed the recommendations of Muhammad (Muhammad LN. Guidelines for repeated measures statistical analysis approaches with basic science research considerations. J Clin Invest. 2023 Jun 1;133(11):e171058. doi: 10.1172/JCI171058. PMID: 37259921; PMCID: PMC10231988.).

ANOVAs with repeated measures like in our case are particularly susceptible to the violation of the assumption of sphericity. As part of the presented ANOVAs we performed Mauchly's test of sphericity. Due to the Mauchly's test of sphericity, the sphericity assumption was violated in all calculations. As usual, we therefore performed an adjustment with the Huynh-Feldt correction was performed. Hence, all data presented (mean square, F, p-values) include these corrections.

We added in the Methods, statistical paragraph, line 149-152:

“Due to the Mauchly's test of sphericity, the sphericity assumption was violated in all ANOVA calculations. Therefore an adjustment with the Huynh-Feldt correction was performed. Hence, all ANOVA data presented (mean square, F, p-values) include these corrections.”

With regard to missing data, a statement has been added in the Methods in lines 147-149. We followed international standard handling missing data (Muhammad, see above, or for instance: “The prevention and handling of the missing data” https://doi.org/10.4097/kjae.2013.64.5.402):

“The rate of missing data was less than 5% per parameter. Based on the assumption that the missingness occurred completely are random, the listwise deletion approach was used to handle the missing data in the ANOVA calculations.”

An additional statement has been added to the discussion section in line 404-405:

 “Furthermore, missing values, although this was rare, could have led to a selection bias”

1.2. Clarification of General Treatment Guidelines: Your study focuses on preventive versus reactive PEG placement in head and neck cancer patients. However, the manuscript lacks a comprehensive description of the general treatment guidelines for head and neck cancer patients at your institution and in Germany. Specifically, PEG is considered a last resort, and nasogastric tubes or TPN are commonly used for acute nutritional management in some regions. You need to clarify why PEG was chosen and why TPN was not considered. This clarification is essential as it directly impacts decisions regarding pre- or post-treatment PEG placement. The rationale behind the clinical decision-making process for preventive versus reactive PEG should be more detailed.

Answer 1.2: This is an important point. We added more information on the clinical decision-making process according to the German treatment guidelines in the method section in line 119-136:

“The decision in favor of a PEG or a nasogastric tube was made in accordance with the German guidelines for the nutrition of patients with different types of HNC. These generally state that artificial feeding, a nasogastric tube or PEG is indicated in the postoperative period. If it is to be expected that oral nutrition will no longer be possible for at least a few weeks, the PEG tube has proven to be safe and effective. If tube feeding (transnasal or transcutaneous) is necessary, the PEG should be preferred to a nasogastric tube if dysphagia is present or expected to persist. In uncomplicated cases, however, feeding with soft or liquid high calorie food under local and systemic analgesia is recommended, provided the patient does not aspirate. If these measures are not sufficient, nutrition must be provided via a PEG or nasogastric tube or parenterally. Prophylactic PEG placement is recommended if intensive radiotherapy or radiochemotherapy is planned (Lee et al. 1998). However, the German guidelines also points out that the available literature shows no significant benefit of prophylactic PEG placement in terms of weight progression, quality of life and oncological outcome (Brown et al. 2017, Axelsson et al. 2017, Silander et al. 2013). Furthermore, it is described that there is no evidence for the correct timing of PEG placement (Deurloo et al. 2001, Beaver et al.). It is also recommended to replace a nasogastric tube with a PEG if dysphagia and especially aspiration continue to occur after the healing phase and intensive swallowing training.”

References:

Deutsche Gesellschaft für Mund-Kiefer- und Gesichtschirurgie e.V. (DGMKG) (2021, 02/03/2021). "S3-Leitlinie Diagnostik und Therapie des Mundhöhlenkarzinoms."  3.0. Retrieved 22/12/2023, from https://register.awmf.org/de/leitlinien/detail/007-100OL.

Deutsche Gesellschaft für Hals-Nasen-Ohren-Heilkunde; Kopf- und Hals-Chirurgie e.V. (DGHNO-KHC) (2023, 14/11/2023). "S3-Leitlinie Konsultationsfassung: Diagnostik, Therapie, Prävention und Nachsorge des Oro- und Hypopharynxkarzinoms."  1.01. 22/12/2023, from Deutsche Gesellschaft für Hals-Nasen-Ohren-Heilkunde, Kopf- und Hals-Chirurgie e.V. (DGHNO-KHC).

Deutsche Gesellschaft für Hals-Nasen-Ohren-Heilkunde; Kopf- und Hals-Chirurgie e.V. (DGHNO-KHC) (2019, 31.01.2019). "S3-Leitlinie Diagnostik, Therapie und Nachsorge des Larynxkarzinoms."  1.1. 22/12/2023, from https://register.awmf.org/de/leitlinien/detail/017-076OL.

Lee, J.H., et al., Prophylactic gastrostomy tubes in patients undergoing intensive irradiation for cancer of the head and neck. Arch Otolaryngol Head Neck Surg, 1998. 124(8): p. 871-5. http://www.ncbi.nlm.nih.gov/pubmed/9708712

Brown T, Banks M, Hughes B, Lin C, Kenny L, Bauer J. Impact of early prophylactic feeding on long term tube dependency outcomes in patients with head and neck cancer. Oral Oncol. 2017;72:17-25. URL: https://pubmed.ncbi.nlm.nih.gov/28797454/ 737.

Axelsson L, Silander E, Nyman J, Bove M, Johansson L, Hammerlid E. Effect of prophylactic percutaneous endoscopic gastrostomy tube on swallowing in advanced head and neck cancer: A randomized controlled study. Head Neck. 2017;39(5):908-915. URL: https://pubmed.ncbi.nlm.nih.gov/28152219/ 738.

Silander E, Jacobsson I, Bertéus-Forslund H, Hammerlid E. Energy intake and sources of nutritional support in patients with head and neck cancer--a randomised longitudinal study. Eur J Clin Nutr. 2013;67(1):47-52. URL: https://pubmed.ncbi.nlm.nih.gov/23169469

Deurloo, E.E., et al., Percutaneous radiological gastrostomy in patients with head and neck cancer. European Journal of Surgical Oncology, 2001. 27(1): p. 94-7. http://www.ncbi.nlm.nih.gov/pubmed/11237498

Beaver, M.E., et al., Percutaneous fluoroscopic gastrostomy tube placement in patients with head and neck cancer. Arch Otolaryngol Head Neck Surg, 1998. 124(10): p. 1141-4. http://www.ncbi.nlm.nih.gov/pubmed/9776193

1.3. Impact of PEG Placement on QOL: Your study suggests that PEG placement may lead to a decline in QOL. However, multiple studies have reported that proper nutritional management is crucial for maintaining QOL. This point requires a more detailed discussion and the introduction of counterevidence. Was the decline in QOL due to bias, issues in the treatment plan, or other factors?

Answer 1.3: Thank you for this comment. There are some reports in the literature that appropriate nutritional management is crucial for maintaining quality of life. However, a direct comparison seems difficult or impossible to us, as such current studies often have a prospective study design and a subjective PEG assessment is carried out (Hausmann et al. 2020, Löser et al. 2022). A subjective PEG assessment is due to our retrospective study design not possible. Therefore, only the assumption of a poorer quality of life with PEG due to the greater weight loss test was made. However, we have added a supplementary sentence in the discussion section in lines 105-114 on the quality of life.

 “Low weight loss also had a positive effect on quality of life. As patients with oral nutrition showed the least weight loss in this study, it can be assumed that these patients had the highest quality of life. Prevost et al. also found that a better nutritional status correlates with a better quality of life (Prevost et al. 2014).”

Literature:

Hausmann J, Kubesch A, Goettlich CM, Rey J, Wächtershäuser A, Bojunga J, Blumenstein I. Quality of life of patients with head and neck cancer after prophylactic percutaneous-gastrostomy. Eur J Clin Nutr. 2020 Apr;74(4):565-572. doi: 10.1038/s41430-019-0499-5. Epub 2019 Sep 30. PMID: 31570758.

Löser A, Avanesov M, Thieme A, Gargioni E, Baehr A, Hintelmann K, Tribius S, Krüll A, Petersen C. Nutritional Status Impacts Quality of Life in Head and Neck Cancer Patients Undergoing (Chemo)Radiotherapy: Results from the Prospective HEADNUT Trial. Nutr Cancer. 2022;74(8):2887-2895. doi: 10.1080/01635581.2022.2042571. Epub 2022 Feb 25. PMID: 35209777.

Prevost V, Joubert C, Heutte N, Babin E. 2014. Assessment of nutritional status and quality of life in patients treated for head and neck cancer. European Annals of Otorhinolaryngology-Head and Neck Diseases, 131 (2):113-120.

1.4. Detailed Description of the Nutritional Screening Process: Currently, the manuscript lacks a detailed description of the nutritional screening process, including the specific procedures and criteria for KPS evaluation. It is important to clarify who conducted the KPS evaluations, at what stage, and whether these evaluations were consistent across different patients.

Answer 1.4: We added more information about the description of the nutritional screening process in the method section in line 105-114.

“One of the most important and simplest ways to implement a nutritional process is anthropometry. This primarily involved measuring height and weight, and using this data to calculate the body mass index (BMI). These information on height, weight and nutritional characteristics including patient, tumor and treatment characteristics were obtained from the tumor board and operative reports. Letters from departments of ENT and Radiation Oncology contained important information about follow-up at the previously defined times (Z0-Z3).

It is also useful to assess nutritional status in combination with performance status and quality of life. The Karnofsky performance status was used for classification. The Karnofsky performance status was also taken from the tumor board reports written by the treating physician.”

1.5. Impact of Alcohol and Nicotine Use: The manuscript briefly touches upon the impact of alcohol and nicotine use on nutritional status and treatment outcomes. However, further research is needed on this topic. Could you provide more detailed data on how these factors affected patient outcomes and whether subgroup analyses were performed, such as for patients who smoked enough to develop COPD or drank enough to develop alcohol-related liver disease?

Answer 1.5: The impact of alcohol and nicotine use on patient outcome is an important factor, but was not the main focus of this study. The retrospective data collection did not provide sufficient information on COPD and alcohol-related liver disease. Due to a lack of information on other diseases, no subgroup analyses were carried out.

Mussab Kouka and Orlando Guntinas-Lichius

for all authors

Jena, 03-Sep-2023

Reviewer 2 Report

Comments and Suggestions for Authors

I have reviewed this interesting retrospective study of 289 patients with head and neck cancer regarding the role of percutaneous endoscopic gastrostomy (PEG).

Methods, item 2.2: Inclusion and exclusion criteria should be further described. 

Which were the criteria for indication PEG in this cohort?

How many patients with indication for PEG refused it? Did the outcome of those patients could impact on your findings among the patients who did not get PEG?

Were not albumin and other laboratorial criteria considered in this study?

Was it appropriate to include thyroid cancer patients in this study?

Is p16 study performed for all head and neck cancer patients regardless the primary site? Did it include thyroid and salivary gland tumors?

Would not be more appropriate include only patients under clinical staging III and IV in this study?

The patients were enrolled from the period between 2017 and 2019 for guaranteeing a sufficient follow up time. Which was the impact of PEG placement of the oncological outcome?

The results found in this retrospective study are confirmatory to the current literature.

Author Response

Point-by-Point Rebuttal Letter - Reviewer #2

Manuscript ID cancers-3184364

Role of percutaneous endoscopic gastrostomy for the nutrition of head and neck cancer patients before and up to 6 months after cancer treatment

We thank the reviewers for their helpful comments. We answer here all comments and queries point-by-point.

Reviewer #2

I have reviewed this interesting retrospective study of 289 patients with head and neck cancer regarding the role of percutaneous endoscopic gastrostomy (PEG). Methods, item 2.2: Inclusion and exclusion criteria should be further described.

2.1. Which were the criteria for indication PEG in this cohort?

Answer 2.1: We added more information about the criteria for indication PEG in the method section in line 99-101.

“PEG insertion was primarily indicated in cases of low initial weight, severe restrictions on oral nutrition and before planned chemotherapy or radiochemotherapy treatment.”

2.2. How many patients with indication for PEG refused it? Did the outcome of those patients could impact on your findings among the patients who did not get PEG?

Answer 2.2:  Data on the number of rejected PEGs was not collected. As a retrospective study, the decision could not be determined retrospectively. This typical limitation of a retrospective design was already addressed in the limitation paragraph in the Discussion.

2.3. Were not albumin and other laboratorial criteria considered in this study?

Answer 2.3: This retrospective study focused on patient, tumor and symptom characteristics. Laboratory parameters such as albumin were not collected. We added this limitation and references in the Discussion, limitation section, lines 415-418:

“Although nutritional risk screening scores typically do not include blood values due to high variability [49], some factors like albumin seem to be surrogate markers for the nutritional status of a head and neck cancer patient [50]. Hence, it would be worthwhile to include metabolic blood values in further investigations.”

2.4. Was it appropriate to include thyroid cancer patients in this study?

Answer 2.4: Although the proportion of thyroid tumors was low, they were included in this study for the sake of completeness. Due to TNM, thyroid cancer is head and neck cancer. In addition, this study also focused on symptom-related analysis. Due to their location, thyroid cancers already cause pronounced swallowing difficulties preoperatively and should therefore be taken into account.

2.5. Is p16 study performed for all head and neck cancer patients regardless the primary site? Did it include thyroid and salivary gland tumors?

Answer 2.5: Thank you for this hint. Yes, the HPV 16 status was determined for all typical head and neck cancer subsites, i.e. not only for oropharyngeal cancer. This is routine in the pathology department. No, this was not done for thyroid and salivary gland cancer. Hence, the description in Table 1 had to be corrected.

2.6. Would not be more appropriate include only patients under clinical staging III and IV in this study? (also addressed by the academic editor)

Answer 2.6: We present real world data, i.e. also some patients with stage I/II cancer received nutritional support. This was the topic of this study, i.e. independent of the stage. Furthermore, we screened the literature for other study reporting PEG use in head and neck cancer patients. For instance, in Bäck et al., 14% of the n=292 patients had a UICC stage I/II (Acta Otolaryngol . 2014 Jul;134(7):760-7. doi: 10.3109/00016489.2014.895040. Epub 2014 May 5). In Ruohoalho et al, 12% were in stage I/II (Eur Arch Otorhinolaryngol . 2017 Nov;274(11):3971-3976. doi: 10.1007/s00405-017-4732-3.). In Raynor et al. is was 4% (Otolaryngol Head Neck Surg . 1999 Apr;120(4):479-82. doi: 10.1053/hn.1999.v120.a91408.). Hence, although most patients needing a PEG are in advanced stage, is not usual that also stage I/II patients are needing a PEG.

2.7. The patients were enrolled from the period between 2017 and 2019 for guaranteeing a sufficient follow up time. Which was the impact of PEG placement of the oncological outcome?

Answer 2.7: The impact of PEG use on oncological outcomes such as overall survival is an important factor, but was not the focus of this study. We plan to analyze the impact on outcome in a second study. A sentence was added in the Discussion in lines 407-408:

We are currently conducting a next study to investigate the impact of PEG use on survival.”

The reference list was also updated (without track changes mode!) according to the statements in this rebuttal.

Mussab Kouka and Orlando Guntinas-Lichius

for all authors

Jena, 03-Sep-2023

Round 2

Reviewer 1 Report

Comments and Suggestions for Authors

You answered the questions I asked about the statistics.

 Your answers about the current status of PEG expansion for HNSCC patients in Germany exceeded my expectations and are reflected in the paper. My other wishes were also fulfilled.

Reviewer 2 Report

Comments and Suggestions for Authors

The study is interesting and I appreciate the authors' efforts, however, some limitations were kept in the last version. The study was retrospective ajust confirmed the current literature.